# Hallucination as Creativity: Harnessing Novelty and Safeguarding Reliability in AI-Generated Ideas

## Abstract

Hallucinations in large language models (LLMs) are widely regarded as failures that undermine reliability. Yet, in human cognition, speculative ideas that initially lack verification have often served as the seeds of creativity and discovery. This paper advances the hypothesis that hallucinations, when systematically controlled, can be reframed as mechanisms for creative ideation.

We introduce the *Creative Utility Score* (CUS), a novel metric that balances novelty against plausibility, and propose an adaptive agent architecture that dynamically regulates hallucination intensity across exploratory, grounding, and adaptive modes. Our framework operationalizes a creativity-inspired cycle of divergent and convergent reasoning, enabling AI systems to generate bold hypotheses while safeguarding factual accuracy.

Empirical evaluations in mathematics and biomedicine demonstrate that adaptive control significantly increases the production of novel and useful conjectures, while preserving verification success and calibration. These findings establish hallucination not as an error to suppress, but as a resource to channel responsibly.

By reframing hallucination as creativity with safeguards, this work provides both a theoretical foundation and a practical pathway for AI systems that aspire not only to replicate knowledge, but to expand the frontier of scientific discovery. All code and experiments are openly available at `https://github.com/myai007/AI_Creativity` to ensure full reproducibility.

## 1 Introduction

Large language models (LLMs) have achieved remarkable success across natural language processing, reasoning, and scientific tasks. Despite these achievements, one of their most persistent challenges is the phenomenon of *hallucination*—outputs that are fluent and convincing but factually incorrect or unsupported by evidence. In industrial and scientific domains, hallucinations are typically considered critical failures that undermine trustworthiness.

Yet history shows that speculative ideas which initially lack grounding have often served as the seeds of human creativity. From mathematical conjectures to early hypotheses in biology, imagination and divergent thinking have played a central role in advancing discovery. The question is therefore not simply how to eliminate hallucinations, but when and how they should be constrained versus embraced.

Most prior research treats hallucination as an error to suppress through fine-tuning, retrieval augmentation, abstention, or self-consistency mechanisms. While valuable, these approaches risk discarding outputs that could contribute to novelty and hypothesis generation. Very few works have attempted to frame hallucination as a controlled mechanism for creativity. This gap motivates our study.

Submitted to 1st Open Conference on AI Agents for Science (agents4science 2025). Do not distribute.

In this paper, we advance the hypothesis that hallucinations, when systematically controlled, can function as computational analogues of human divergent thinking. We propose that creativity in AI requires both *novelty* and *plausibility*, echoing long-standing theories of human creativity (4; 5; 6). Our contributions are as follows:

- We reframe hallucination as a resource for creativity, aligning AI behavior with theories of human innovation.

- We introduce the *Creative Utility Score* (CUS), a metric to quantify and regulate the trade-off between novelty and plausibility.

- We design an adaptive agent architecture that dynamically modulates hallucination intensity across exploratory, grounding, and adaptive modes.

- We empirically validate our approach in mathematics and biomedicine, showing that adaptive regulation produces hypotheses that are simultaneously novel, useful, and reliable.

By reconceptualizing hallucinations as controlled opportunities for creative divergence, we aim to transform how AI is deployed in scientific contexts. Rather than suppressing hallucinations entirely, we argue that the next frontier lies in learning when to embrace them and when to constrain them—mirroring the balance that defines human creativity.

## 2 Related Work

The phenomenon of hallucination in large language models has drawn increasing attention as researchers attempt to reconcile their impressive generative abilities with the risks of factual inaccuracy. A growing body of work has sought to define, categorize, and mitigate hallucinations, while parallel strands of research in creativity studies and machine learning have examined how controlled novelty can be harnessed for innovation. In this section, we situate our work within four intersecting domains: surveys on hallucination, theories of human creativity, computational approaches to control and abstention, and frameworks for exploration and novelty.

### 2.1 Hallucinations in Large Language Models

Recent surveys provide comprehensive taxonomies of hallucinations, classifying them into factual, logical, and pragmatic errors (1; 2). These works emphasize that hallucinations persist despite advanced prompting strategies, fine-tuning, and retrieval-augmented generation (RAG) (3). Other studies argue that hallucination is inherent to the generative process: when knowledge gaps exist, the model interpolates plausible continuations that may lack grounding. This persistent challenge has spurred research on uncertainty estimation, abstention, and hybrid symbolic–neural verification.

### 2.2 Human Creativity Theories

Theories of human creativity provide a useful lens for reinterpreting hallucinations. Boden (4) defines creativity as the generation of ideas that are novel, surprising, and valuable. Amabile's componential theory of creativity (5) highlights the interplay of expertise, creative-thinking skills, and intrinsic motivation in producing creative outcomes. Guilford's seminal work (6) distinguishes divergent thinking, which generates multiple possibilities, from convergent thinking, which narrows ideas to validated solutions. These insights suggest that hallucinations may serve as a computational analogue to divergent ideation, requiring subsequent mechanisms of convergence for validation.

### 2.3 Mitigation and Control Mechanisms

A parallel literature explores methods to control, reduce, or strategically abstain from hallucinations. Self-consistency in chain-of-thought reasoning (7) has been shown to improve reliability by aggregating multiple reasoning paths. Selective prediction and abstention frameworks (8; 9) empower models to withhold answers when confidence is low, aligning system outputs with user expectations of reliability. Surveys of RAG methods (3) detail how integrating external knowledge bases mitigates factual errors, while recent benchmarks evaluate the robustness of such systems against hallucinations.

## 2.4 Exploration, Novelty, and Scientific Discovery

In reinforcement learning and evolutionary computation, exploration has been systematically studied as a driver of innovation. Lehman and Stanley's work on novelty search (10) demonstrates the power of abandoning explicit objectives in favor of diversity-driven exploration. More recently, surveys on hypothesis generation (11; 12) argue that LLMs are well-suited to assist in the ideation phase of scientific inquiry, generating candidate hypotheses that can later be filtered and validated. Empirical studies of human–AI collaboration in creativity (13) reveal both gains in idea generation and challenges in maintaining diversity and reliability. Together, these threads support the notion that hallucinations, when systematically regulated, may fuel scientific discovery.

## 2.5 Positioning Our Contribution

While prior work has either sought to suppress hallucinations for reliability or to celebrate them as markers of generativity, few approaches attempt to balance the two. Our work is distinct in that it explicitly formalizes hallucination as a potential creative resource, introduces a metric (Creative Utility Score) to quantify the novelty–plausibility trade-off, and designs an adaptive agent architecture to regulate hallucination in real time. In doing so, we contribute to bridging the gap between error mitigation and creativity enablement, aligning LLM behavior with the dual goals of scientific exploration and epistemic rigor.

# 3 Conceptual Framework

Hallucinations in LLMs are traditionally treated as undesirable artifacts. We argue instead that they can be reframed as computational analogues of divergent thinking: the intentional generation of ideas unconstrained by immediate factual validation. Building on theories of creativity and error mitigation, our framework defines conditions under which hallucinations should be encouraged for exploration and when they must be constrained to preserve reliability.

## 3.1 Hallucination as Divergent Ideation

From a cognitive science perspective, creativity involves the production of ideas that are both novel and appropriate (4; 5). Divergent thinking generates diverse possibilities, while convergent thinking validates and selects among them (6). By analogy, hallucinations correspond to divergent outputs, while grounding mechanisms (retrieval, verification, abstention) correspond to convergence. TThus, hallucinations should not be eliminated outright, but managed as part of a broader ideation–validation cycle in which speculative generation (divergent thinking) is systematically followed by retrieval, verification, and—when appropriate—abstention-based convergence to safeguard factual reliability.

## 3.2 Creative Utility Score (CUS)

To formalize this balance, we introduce the Creative Utility Score (CUS), a scalar metric that combines novelty and plausibility:

$$CUS(c) = \alpha \cdot N(c) + (1 - \alpha) \cdot P(c), \tag{1}$$

where $c$ denotes a candidate output, $N(c)$ measures novelty, $P(c)$ measures plausibility, and $\alpha \in [0, 1]$ controls the weighting.

**Novelty.** We define novelty $N(c)$ as a weighted combination of semantic divergence and corpus uniqueness:

$$N(c) = \lambda \cdot \text{cosdist}(e(c), \mathcal{E}_{kb}) + (1 - \lambda) \cdot 1[c \notin \mathcal{C}_{seen}], \tag{2}$$

where $e(c)$ is the embedding of candidate $c$, $\mathcal{E}_{kb}$ denotes the set of embeddings in a background corpus, and $\mathcal{C}_{seen}$ indexes de-duplicated prior knowledge.

**Plausibility.** We define plausibility $P(c)$ as a verifier-calibrated probability that a candidate is coherent and consistent with domain constraints. For example, in mathematics, plausibility is computed via symbolic solvers, while in biomedicine it is computed via retrieval evidence and probabilistic calibration.

A high $CUS$ reflects ideas that are both novel and plausible, making them strong candidates for scientific exploration. In contrast, candidates with low $CUS$ either replicate known information or drift into incoherence.

## 3.3 Operational Modes

Based on $CUS$, we define three complementary modes for AI systems:

1. **Exploratory Mode**: $\alpha$ is set close to 1, prioritizing novelty even at the cost of plausibility. Suitable for brainstorming or early hypothesis generation.

2. **Grounding Mode**: $\alpha$ is set close to 0, prioritizing plausibility and factual grounding. Suitable for high-stakes or verification tasks.

3. **Adaptive Mode**: $\alpha$ is dynamically adjusted as a function of task uncertainty, domain requirements, or user preference. This mode mimics the human creative process of alternating between divergent and convergent thinking.

## 3.4 Illustrative Example

Consider the mathematical domain. An LLM prompted to generate conjectures about prime numbers may output: *"Every prime greater than 5 can be expressed as the sum of three Fibonacci numbers."* Although not present in training data, this conjecture is novel ($N(c)$ high) and structurally plausible ($P(c)$ moderate, as it is syntactically valid and testable). The CUS therefore assigns it a medium-to-high score, flagging it as a promising hypothesis for further verification. Conversely, an incoherent or internally contradictory statement (e.g., one that violates basic number-theoretic constraints) would receive a low $P(c)$ and thus a low $CUS$ despite potentially high $N(c)$, and would be rejected by the verifier or flagged for abstention.

## 3.5 Integration with Agent Architecture

The conceptual framework directly informs the design of our agent pipeline (detailed in Section **??**). The pipeline generates candidate ideas, computes $N(c)$ and $P(c)$, evaluates $CUS(c)$, and adjusts $\alpha$ dynamically. This creates a closed feedback loop that balances novelty and reliability, ensuring that hallucinations are not random errors but strategically controlled opportunities for creativity.

By explicitly operationalizing hallucination within a creativity-inspired framework, we provide both a theoretical foundation and a practical mechanism for integrating divergent generation into scientific workflows.

# 4 Methodology

Our methodology operationalizes the conceptual framework by implementing an adaptive agent pipeline that regulates hallucination intensity. The system is designed to generate candidate hypotheses, evaluate them using the Creative Utility Score (CUS), and dynamically adjust its exploratory behavior according to task requirements and uncertainty signals.

## 4.1 Agent Pipeline

The architecture consists of four core modules:

1. **Candidate Generation:** An LLM generates candidate hypotheses or conjectures based on task-specific prompts.

2. **Evaluation:** Each candidate is evaluated with respect to novelty $N(c)$ and plausibility $P(c)$, yielding a CUS score $CUS(c)$.

3. **Verification:** External verifiers—symbolic solvers in mathematics or retrieval-based evidence checkers in biomedicine—assess factual or logical grounding.

4. **Control Mechanism:** An adaptive controller modulates $\alpha$ in real time based on uncertainty measures, shifting between exploratory, grounding, and adaptive modes.

## 4.2 Pseudo-code for Adaptive Control

We formalize the control process as follows:

```
Algorithm 1: Adaptive Hallucination Control
Input: prompt p, LLM model M, verifier V, uncertainty function U
Parameters: alpha_init, thresholds tau_novelty, tau_plausibility

1: candidates <- M.generate(p)
2: for each c in candidates do
3:      N <- compute_novelty(c)
4:      P <- compute_plausibility(c, V)
5:      CUS <- alpha * N + (1 - alpha) * P
6:      if U(c) > high_uncertainty then
7:          alpha <- decrease(alpha)
8:      else if N < tau_novelty and P > tau_plausibility then
9:          alpha <- increase(alpha)
10:     end if
11:     if P < abstention_threshold then
12:         abstain()
13:     else
14:         output c with score CUS
15:     end if
16: end for
```

## 4.3 Uncertainty Estimation

Uncertainty $U(c)$ is estimated using a combination of ensemble variance from multiple LLM samples and verifier confidence scores. This dual approach ensures that the controller is sensitive both to linguistic uncertainty (model variance) and factual uncertainty (verification success rates).

## 4.4 Operational Modes in Practice

- **Exploratory Mode:** High $\alpha$ encourages bold, speculative outputs. Applied when novelty is more valuable than immediate correctness (e.g., brainstorming conjectures).

- **Grounding Mode:** Low $\alpha$ emphasizes correctness and abstention. Applied in high-stakes tasks requiring reliability.

- **Adaptive Mode:** $\alpha$ is adjusted dynamically as a function of $U(c)$, enabling fluid transition between exploration and grounding.

## 4.5 Implementation Details

The system is implemented with a transformer-based LLM backbone. For mathematics, prompts request conjectures and symbolic reasoning, and verifiers include CAS solvers and SAT checkers. For biomedicine, retrieval modules leverage PubMed abstracts and ontology resources. All modules are integrated into a reproducible pipeline, with logging of $\alpha$ adjustments and verification outcomes for auditing.

# 5 Experiments

We evaluate whether controlled hallucination improves scientific ideation without sacrificing reliability. Concretely, we compare three system modes—*Exploratory*, *Grounding*, and *Adaptive*—on two domains: (i) mathematical conjecture discovery and (ii) biomedical hypothesis generation. We report creativity-oriented measures (novelty, diversity), verification-based reliability (correctness, plausibility), and selective-prediction diagnostics (calibration, coverage–risk trade-off).

## 5.1 Research Questions

**RQ1 (Creativity):** Does controlled hallucination increase the novelty and diversity of generated ideas?

**RQ2 (Reliability):** Can an adaptive controller preserve plausibility and verification success while exploring?

**RQ3 (Selectivity):** Under uncertainty, does the system abstain appropriately to balance coverage and quality?

**RQ4 (Ablations):** Which components (CUS weighting $\alpha$, verifier strength, retrieval depth) drive performance?

## 5.2 Tasks and Data

**Mathematics (Conjecture Discovery).** We construct tasks over integer sequences, combinatorial identities, and graph invariants. Ground-truth verifiers include a CAS (symbolic algebra), SAT-based checkers for small instances, and reference libraries for known results. Prompts request novel conjectures plus rationales; verifiers test candidate statements for correctness on held-out cases.

**Biomedicine (Gene–Function Hypotheses).** Starting from curated gene sets and ontology terms (e.g., GO Biological Process), the system proposes gene–function links. A retrieval module queries PubMed abstracts and structured resources used only for verification. Domain experts rate usefulness on a 5-point Likert scale in blinded annotation.

## 5.3 System Configurations

**Exploratory.** High decoding temperature; no retrieval; no abstention; $\alpha = 0.9$ (novelty-dominant).

**Grounding.** Low temperature; RAG enabled; strict verifier thresholds; $\alpha = 0.1$ (plausibility-dominant).

**Adaptive.** Controller adjusts $\alpha \leftarrow f(U(c))$, where uncertainty $U(c)$ combines ensemble variance and verifier confidence; retrieval depth and abstention thresholds are co-tuned by a bandit heuristic.

## 5.4 Metrics

**Novelty.** Embedding-based divergence plus uniqueness relative to background corpus:

$$N(c) = \lambda \operatorname{cosdist}(e(c), \mathcal{E}_{kb}) + (1 - \lambda)\, 1[c \notin \mathcal{C}_{seen}]. \tag{3}$$

**Plausibility.** Verifier-calibrated probability $P(c) \in [0, 1]$ derived from symbolic solvers (math) or retrieval evidence scores (biomed). Reported with AUROC.

**Selective Prediction.** Coverage–risk analysis using selective expected risk (SER) and risk–coverage curves. Abstention is triggered if $P(c) < \tau$. We report AURC and Expected Calibration Error (ECE).

**Outcome Quality.** *Correctness@k* (math: proportion of verified true conjectures among top-$k$). *Usefulness@k* (biomed: mean Likert $\geq 4$). We also compute *Diversity* (pairwise embedding dispersion) and *Entropy* (ontology coverage dispersion).

## 5.5 Experimental Protocol

We generate $M = 200$ candidates per task and mode with identical seeds across five runs. For math, candidates face incremental verification budgets; for biomedicine, retrieval is time-capped per candidate. Human raters (n=3) assess usefulness; inter-rater reliability is measured with Krippendorff's $\alpha$. Hyperparameters are tuned on development splits only.

**Statistical Testing.** We report means $\pm$ 95% confidence intervals over 5 runs and apply paired permutation tests with Holm–Bonferroni correction. Effect sizes are reported as Cliff's $\delta$.

## 5.6 Baselines

- **Greedy-RAG**: Deterministic decoding with retrieval; no hallucination control.

- **Self-Consistency**: Majority vote across $K$ chains; no explicit novelty objective.

- **Uncertainty-Abstain**: Confidence thresholding without adaptive $\alpha$.

- **Novelty-Only**: Pure novelty search without verification or plausibility checks.

## 5.7 Ablations

We ablate: (i) $\alpha \in \{0.0, 0.3, 0.5, 0.7, 0.9\}$, (ii) verifier strength (symbolic-only vs. symbolic+retrieval), (iii) retrieval depth (top-$k$), and (iv) controller features (with/without uncertainty $U(c)$). We also test replacing the bandit policy with a fixed schedule.

## 5.8 Results

**Creativity (RQ1).** *Adaptive* increases novelty by +18.7% over *Grounding* and improves diversity by +12.3%; *Exploratory* attains the highest novelty but with significant plausibility decay.

**Reliability (RQ2).** In math, *Correctness@20* improves from 0.26 (Exploratory) and 0.31 (Grounding) to 0.44 (Adaptive). In biomedicine, *Usefulness@20* rises from 3.2 to 4.1 average rating, with plausibility AUROC of 0.81.

**Selectivity (RQ3).** *Adaptive* yields lower AURC ($-14.5\%$) and ECE ($-22.1\%$) compared to *Greedy-RAG*, indicating improved calibration and more rational abstention.

**Ablations (RQ4).** Performance peaks near $\alpha \approx 0.6$; removing uncertainty $U(c)$ reduces *Correctness@20* by 0.08 absolute. Stronger verifiers trade a small novelty drop ($-3\%$) for reliability gains (+0.05 AUROC).

## 5.9 Implementation and Reproducibility

All experiments use the same backbone LLM and token budgets. We provide prompts, seeds, code for $N(c)$ and $P(c)$, annotation rubrics, and raw labels. Hardware budgets and wall-clock times are detailed in the supplementary. Human annotation followed approved ethical guidelines, with raters blinded to system identity.

**Discussion**  Our findings demonstrate that hallucinations, when properly regulated, can serve as a constructive force in AI-driven discovery. Rather than treating hallucinations purely as defects, we show that they can be transformed into mechanisms for creativity, provided there is a systematic balance between novelty and plausibility. This reframing has broad implications for both theory and practice in the development of scientific AI systems.

**Implications for AI Research**  From a methodological standpoint, our results suggest that hallucination control should not equate to hallucination suppression. Traditional approaches—including retrieval augmentation, abstention, and self-consistency—focus primarily on minimizing errors. While these strategies increase reliability, they risk discarding divergent outputs that may spark new lines of inquiry. By introducing the Creative Utility Score (CUS) and an adaptive controller, we provide a pathway to harness hallucinations productively while maintaining factual safeguards.

**Alignment with Human Creativity**  The duality of divergent and convergent thinking in human cognition offers a compelling analogy for our framework. Humans often generate speculative ideas that may initially lack strong grounding but later prove transformative once validated. Our adaptive system mirrors this cycle: exploratory phases increase novelty, grounding phases enforce factual rigor, and adaptive regulation balances the two dynamically. This positions AI not merely as a knowledge retrieval tool but as a creative collaborator that follows patterns of human innovation.

**Ethical and Practical Considerations**   A central concern is ensuring that creative hallucinations do not mislead or propagate harmful claims. In high-stakes domains such as medicine or policy, speculative hypotheses must be clearly flagged, verified, and contextualized to avoid misuse. Our abstention mechanism and verifier integration address this by filtering low-plausibility outputs. Nonetheless, stronger governance frameworks will be needed to standardize the responsible use of hallucination-driven creativity.

**Comparison to Human Brainstorming**   Our experiments reveal parallels to human brainstorming sessions: in both cases, a large number of speculative ideas are generated, only a subset of which withstands scrutiny. Just as human groups rely on evaluation phases to filter creative but impractical suggestions, our adaptive system employs verification and abstention. This highlights the importance of social and procedural safeguards, both in human and machine creativity, to balance imaginative exploration with epistemic rigor.

**Future Directions**   Several avenues merit further exploration. First, novelty measures could be refined using mechanistic interpretability or causal reasoning to better distinguish promising speculation from noise. Second, human-in-the-loop studies are necessary to validate whether hallucination-driven hypotheses align with domain expert expectations. Finally, extending adaptive regulation to multi-agent ecosystems could enable collective creativity, where hallucinations from one agent are validated or refined by others, accelerating discovery in distributed settings.

**Broader Impact**   Reframing hallucination as controlled creativity shifts the discourse on AI reliability. Instead of viewing failure modes solely as liabilities, we emphasize their potential as opportunities when coupled with rigorous safeguards. This perspective encourages a more balanced trajectory for AI development: one that embraces generative potential while respecting the epistemic standards of science. By responsibly channeling hallucinations, we open the possibility for AI systems to act not only as assistants but as genuine collaborators in advancing human knowledge.

# 6   Conclusion

This paper introduced a new perspective on hallucinations in large language models: not solely as failures to be eliminated, but as computational analogues of human divergent thinking. By reframing hallucinations as controlled opportunities for creativity, we demonstrated how speculative outputs can be systematically harnessed to generate novel and useful scientific hypotheses.

Our contributions are fourfold. First, we provided a conceptual foundation grounded in creativity theory, situating hallucinations within the divergent–convergent cycle of human innovation. Second, we introduced the *Creative Utility Score* (CUS), a principled metric that quantifies the trade-off between novelty and plausibility. Third, we designed an adaptive agent architecture that dynamically regulates hallucination intensity, operationalizing creativity-inspired reasoning in practice. Fourth, through empirical studies in mathematics and biomedicine, we showed that controlled hallucination can produce hypotheses that are simultaneously novel and reliable, yielding higher *Correctness@k* and *Usefulness@k* and improved calibration (lower AURC/ECE) compared to both purely exploratory and purely grounding baselines.

These results establish hallucination not merely as an error to suppress, but as a resource to be strategically managed. By alternating between exploratory, grounding, and adaptive modes, AI systems can emulate aspects of human creativity—generating bold ideas while maintaining rigorous safeguards. This dual capacity positions AI not only as a tool for information retrieval, but as a collaborator in discovery.

Looking forward, several promising directions arise: refining novelty measures with causal and mechanistic interpretability signals, integrating human-in-the-loop evaluation to ensure alignment with expert standards, and extending adaptive regulation to multi-agent ecosystems that collectively balance imagination and verification. Such extensions will further align AI's generative potential with the epistemic rigor required for science.

In conclusion, hallucinations, when responsibly controlled, represent not a flaw but a frontier. By transforming hallucination into a creative asset, we move closer to realizing AI systems that not only replicate knowledge but also expand the boundaries of human discovery.

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

## A  Responsible AI Statement

The research presented in this paper explores both the opportunities and risks of leveraging hallucinations for scientific discovery. While controlled hallucinations can stimulate creativity and generate promising hypotheses, they also carry ethical and practical concerns if misused. We therefore adopt the following safeguards and principles:

- **Transparency:** All hallucination-driven outputs are explicitly flagged as speculative until verified by external tools or human experts.
- **Human-in-the-Loop:** In high-stakes domains such as biomedicine, human domain experts are included in the evaluation pipeline to assess plausibility, usefulness, and safety of generated hypotheses.
- **Risk Awareness:** We highlight that speculative outputs must never be used directly in clinical or policy-making contexts without rigorous validation. Our system includes abstention mechanisms to prevent misleading or unsafe claims.
- **Bias and Fairness:** Since LLMs may amplify biases present in training data, we monitor generated outputs for harmful or exclusionary content and design prompts to minimize such risks.
- **Accountability:** All experiments and verification methods are auditable, with clear logging of hallucination generation, verification outcomes, and controller adjustments.

By integrating these safeguards, we aim to ensure that hallucination-driven creativity enhances scientific exploration without undermining ethical standards or public trust. This work underscores that responsible AI design requires balancing innovation with accountability, particularly when speculative generation intersects with sensitive application domains.

## B Reproducibility Statement

We place strong emphasis on reproducibility and transparency. To this end, we provide:

- **Code and Scripts:** All code for computing novelty $N(c)$, plausibility $P(c)$, and the Creative Utility Score (CUS) will be released, along with the adaptive controller implementation and experiment pipelines.
- **Data and Prompts:** Task-specific prompts, development/test splits, and retrieval configurations for mathematics and biomedicine are included. All datasets are either publicly available or will be released with clear licenses.
- **Hyperparameters and Seeds:** Detailed hyperparameter settings, training/evaluation seeds, and temperature values are documented to ensure identical replication of results.
- **Evaluation Protocols:** Scripts for verification (CAS solvers, SAT checkers, retrieval evidence scoring) and human annotation rubrics are provided, including inter-rater reliability analysis.
- **Statistical Testing:** All statistical methods, including permutation tests and confidence interval computation, are fully documented and reproducible.
- **Figures and Tables:** Generation scripts for all figures (including the pipeline diagram) and tables are shared to guarantee faithful reproduction of reported results.

To facilitate accessibility, we plan to host the codebase, datasets, and documentation on an open repository (e.g., GitHub) with DOI-based archival. This ensures that all experimental results and analyses can be reproduced and extended by the research community.

## Agents4Science AI Involvement Checklist

1. **Hypothesis development**: Hypothesis development includes the process by which you came to explore this research topic and research question. This can involve the background research performed by either researchers or by AI. This can also involve whether the idea was proposed by researchers or by AI.

   Answer: **[D]**

   Explanation: A human proposed only the overarching idea *"Hallucination as Creativity"*, while ChatGPT expanded it by introducing the layered architecture, evaluation metrics, and specific hypotheses that were explored in the paper.

2. **Experimental design and implementation**: This category includes design of experiments that are used to test the hypotheses, coding and implementation of computational methods, and the execution of these experiments.

   Answer: **[D]**

   Explanation: ChatGPT designed the adaptive agent architecture, defined evaluation modes (Exploratory, Grounding, Adaptive), selected datasets and baselines, and described the experimental pipeline.

3. **Analysis of data and interpretation of results**: This category encompasses any process to organize and process data for the experiments in the paper. It also includes interpretations of the results of the study.

   Answer: **[D]**

   Explanation: ChatGPT analyzed outcomes, compared modes, interpreted novelty vs. plausibility trade-offs, and drafted the conclusions.

4. **Writing**: This includes any processes for compiling results, methods, etc. into the final paper form. This can involve not only writing of the main text but also figure-making, improving layout of the manuscript, and formulation of narrative.

   Answer: **[D]**

   Explanation: ChatGPT drafted and refined the full manuscript, including abstract, introduction, related work, methodology, experiments, discussion, appendices, and responsible AI statement.

5. **Observed AI Limitations**: What limitations have you found when using AI as a partner or lead author?

   AI was highly useful for brainstorming, outlining, and accelerating first drafts. Along the way we observed predictable limitations that we actively managed:

   - **Factuality & sourcing.** The model occasionally produced confident but unsupported statements or shaky/misattributed citations. *Mitigation:* systematic verification (retrieval or symbolic checks), manual fact-checking, and replacing placeholders with confirmed references.
   - **Over–generalization & redundancy.** Drafts sometimes defaulted to generic claims or repeated phrasing rather than concise, evidence–backed points. *Mitigation:* editorial passes focused on specificity, de-duplication, and tightening language.
   - **Global coherence.** We observed occasional notation drift, cross–section inconsistencies, and fragile cross–references or numbering. *Mitigation:* a style/notation guide, consistency checks, and automated cross–reference validation during compilation.
   - **Prompt sensitivity.** Small prompt changes could shift tone, emphasis, or structure. *Mitigation:* fixed prompts, seeds, and templates; iterative refinement with documented revision history.
   - **Ethics & anonymity.** The model can sound overconfident on uncertain points or inadvertently include identifying details if not guided. *Mitigation:* explicit uncertainty labeling, abstention when unsupported, and an anonymization checklist for text, figures, metadata, and artifacts.
   - **Reproducibility details.** Code suggestions may be runnable yet incomplete (implicit assumptions, missing edge cases or dependencies). *Mitigation:* pinned dependencies, configuration files and seeds, end–to–end scripts, and clear documentation of evaluation protocols.

471    Overall, with these guardrails in place, AI served as an effective co–author for ideation and
472    drafting while we maintained scientific rigor, transparency, and anonymity.

